# Investigation on Foamed PP/Nano-CaCO_3_ Composites in a Combined in-Mold Decoration and Microcellular Injection Molding Process

**DOI:** 10.3390/polym12020363

**Published:** 2020-02-07

**Authors:** Kui Yan, Wei Guo, Huajie Mao, Qing Yang, Zhenghua Meng

**Affiliations:** 1School of Automotive Engineering, Wuhan University of Technology, Wuhan 430070, China; ykui_whut@126.com (K.Y.); yqwhut@163.com (Q.Y.); meng@whut.edu.cn (Z.M.); 2Hubei Key Laboratory of Advanced Technology for Automotive Components, Wuhan University of Technology, Wuhan 430070, China; 3Hubei Collaborative Innovation Center for Automotive Components Technology, Wuhan 430070, China; 4Institute of Advanced Materials and Manufacturing Technology, Wuhan University of Technology, Wuhan 430070, China; 5School of Materials Science and Engineering, Wuhan University of Technology, Wuhan 430070, China

**Keywords:** in-mold decoration, microcellular injection molding, mechanical properties, cellular structure, surface quality

## Abstract

A combined in-mold decoration and microcellular injection molding (IMD/MIM) method has been used in this paper. The foamed PP/nano-CaCO_3_ composites were prepared to investigate their mechanical properties, cellular structure, and surface quality. The content of nano-CaCO_3_ varied from 0 to 10 wt %. The results showed that nano-CaCO_3_ acted as a reinforcing phase and nucleating agent, which help to improve the mechanical properties of foamed composites. The cellular structure and mechanical properties were optimum when the nano-CaCO_3_ content was 6 wt %. In the vertical section, the cell size and density of transition layer on the film side was bigger than that on the non-film side. In the parallel section, the cell ratio of length to diameter of transition layer on the film side was smaller than that on the non-film side, and the cell tile angle was larger than that on the non-film side. With nano-CaCO_3_ content increasing, the surface quality showed a trend of decreasing first and then increasing.

## 1. Introduction 

Microcellular injection molding (MIM) is an effective method to achieve weight reduction of plastic products, with some advantages such as high impact strength, high insulation properties and low electrical conductivity [1,2]. However, MIM products have some shortcomings, such as decrease of tensile strength, bad surface quality, and uneven distribution of cells [3,4,5]. Especially in terms of surface quality, the appearance of the part is poor due to the deformation of cells during the filling period [6]. This has largely limited the application field of MIM.

In order to improve the surface quality of MIM products, there are many studies on eliminating surface defects. By combining rapid heat cycling molding (RHCM) technology with MIM, Huang et al. [7] found that surface roughness can be effectively reduced by controlling the mold temperature. RHCM/MIM was used to investigate the transient flow behavior of foam composites and formation mechanism of the surface defects [8,9,10]. Gas counter pressure (GCP) combined with MIM has been used to improve the surface quality of MIM products by controlling bubbles nucleation in the filling stage [11,12,13]. GCP is the way to inject high-pressure gas into the mold cavity before melt injection and keep a constant back pressure during the melt filling process. Cell nucleation and growth are inhibited due to high pressure in the mold cavity, thus improving the surface quality. These two methods are effective ways to improve surface quality, but they all require design and modification of the mold, which increases the cost of the experiment. Microcellular co-injection molding is also a good way to improve the surface quality of foamed products [14,15]. Microcellular co-injection molding is to inject the non-foamed surface layer material before injecting the foamed core layer material into the mold cavity. Since the surface layer is solid, the surface quality of the obtained part is good. A gas-assisted microcellular injection molding was used to improve the surface quality of products with controlling the bubbles nucleation at the filling stage [16]. A thermal insulation film was attached to the mold cavity to improve the surface quality of the foamed parts [17,18,19]. Jungjoo [18] et al. investigated the effect of film thickness on the surface roughness, and it was found that the surface roughness decreased as the film thickness increased. As for improving cell structure and mechanical properties of foamed products, adding inorganic filler is a commonly used method. Huang investigated the cell structure of the foamed nano-CaCO3/PP in a batch process, and it was found that the foamed PP/nano-CaCO_3_ composite containing 5 wt % nano-CaCO_3_ had the largest cell density [20]. Ding et al. investigated the effects of nano-CaCO_3_ on the foamed PP composites in a batch process [21,22]. They found that with 5 wt % nano-CaCO_3_ addition foamed composites produced optimum cell structure. Mao investigated the content of nano-CaCO_3_ on mechanical properties and the cell structure in different sections, and it was found that with the 6 wt % addition of nano-CaCO_3_, the mechanical properties was optimum [23]. These researchers are more concerned with the effect of fillers on the cell morphology and mechanical properties of foam composites, and there are few papers on the effect of fillers on the surface quality of foamed polymers in MIM. The cell nucleation in the filling stage and the melt strength were all affected due to the addition of filler, which had an impact on the surface quality of parts. 

In this paper, IMD/MIM process was used to improve the surface quality of foamed parts, and different content of nano-CaCO_3_ was added into PP to improve the mechanical properties. The effect of nano-CaCO_3_ content on the cellular structure, mechanical properties, and surface quality was investigated.

## 2. Experimental 

### 2.1. Combined In-Mold Decoration and Microcellular Injection Molding Process

The process of IMD/MIM was shown in Figure 1, the supercritical fluid (SCF) was injected into the barrel through the gas valve, and the SCF and polymer melt were mixed into a single-phase melt under agitation of the screw. The PET film was adhered to the mold cavity surface before the single-phase melt was injected into the mold. The thickness of PET film was 0.2 mm. The single-phase melt was then injected into the mold cavity, the pressure drop induced cell nucleation, and the melt was gradually cooled by the action of the low-temperature mold and film.

As shown in Figure 2, the presence of the film led to a higher temperature on the film side, thus causing cells to shift to the film side. The surface defects were caused by the formation of so-called bubble marks by the deformed bubbles in the filling stage reaching the surface of the part. The higher temperature helped improve the surface quality due to lower melt viscosity and the flat function of film. The lower melt viscosity helped to reduce the friction between the mold and the part, and the higher temperature helped smooth the bubble marks, thus improving the surface quality. 

### 2.2. Preparation of Composites 

The MIM process was shown in Figure 3. During the preparation of composites, all materials were dried by drying oven (101A-1, Guangdi Instrument Equipment Co., Ltd., Shanghai, China). Then materials were put into the mixer (SHR-10, Yiyang plastic Machinery Co., Ltd., Wuhan, China) according to the proportion of each component. Nano-CaCO_3_ contents of composites were 2, 4, 6, 8, 10 wt %, respectively, and 5 wt % PP-g-mah was added as the compatibilizer. The composites were extruded by the twin-screw extruder (SHJ-20, Giant Machinery Co., Ltd., Nanjing, China) and then pelletized by the pelletizer (LQ-20, Giant Machinery Co., Ltd., Nanjing, China). Materials were dried for 24 h at 80 °C. The temperatures of extruder barrel were 170, 180, 180, 180, and 185 °C.

The mean particle diameter of nano-CaCO_3_ (supplied by Changshan Jinxiong Co., Ltd., Changshan, China) was 70 nm. The melt flow index of Polypropylene (K8303, supplied by Sinopec Beijing Yanshan Co., Ltd., Beijing, China) was 2.0 g/10 min. The PP-g-mah (1.2% grafting) was provided by Dongyuan Ziheng Plastics Co., Ltd., Dongguan, China. The industrial N_2_ (99% purity) used as the blowing agent was provided by Wuhan XiangYun Industry Co., Ltd., Wuhan, China. The thermal conductivity of film (provided by China Gwell Machinery Co., Ltd., Shanghai, China) was 0.248 W/m·k.

### 2.3. Foaming Process

As shown in Figure 3, during the foaming process, industrial N_2_ was converted to SCF after being pressurized over 28 MPa by a pressure pump (GBL-200/350, CHN-top Machinery Group Co., Ltd., Beijing, China). The barrel temperature of injection molding machine (HDX50, Haida Plastic Machinery Co., Ltd., Ningbo, China) was 180, 190, 190, and 190 °C. The injection pressure of SCF-N_2_ was controlled by microcellular foaming console (CHN-top Machinery Group Co., Ltd., Beijing, China). The SCF-N_2_ (15 MPa) was injected into the barrel when the composites were plasticized. The injection time of SCF-N_2_ was 2 s and the content of SCF-N_2_ was 0.5%. The composites and SCF-N_2_ turned to a single-phase melt under the mixing function of screw. The injection pressure of injection machine was 80 MPa and the back pressure was 10 MPa. The PET film was attached to the mold cavity surface before melt injection. The composites began to foam after the melt was injected into the mold due to the large pressure drop, and the samples were obtained after cooling for 30 s.

### 2.4. Characterizations

The electromechanical universal test machine (MTS SYSTEMS CMT6104, Eden Prairie, MN, USA) was used to measure the tensile properties and flexural properties. The tensile test method is ISO 527-1:1993 with a crosshead speed of 50 mm/min. The flexural test method is ISO 178:2001 with a speed of 2 mm/min. The impact strength was measured using an impact tester (XJUD-5.5, Chengde Jinjian Testing Instrument Co., Ltd., Chengde, China) according to ISO 180:2000. The average valves of the five samples were taken as the results. The value obtained by dividing the strength by the density is defined as the specific strength.

A JSM-IT300 (JEOL Ltd., Tokyo, Japan) scanning electron microscope (SEM) was used to characterize cell structure and distribution. The flexural samples were immersed in liquid nitrogen for 3 h to be broken, and the fractured surface was coated with a gold layer before SEM observation. As shown in Figure 4, the parallel and vertical section (10 × 4 mm^2^) were taken from the flexural sample to observe the cellular structure, and the middle section of the flexural sample was selected to observe the surface quality.

The average cell diameter could be calculated by the following equation
(1)$$D=\frac{\sum_{i=1}^{n}d_{i}}{n}$$
where $d_{i}\;$is the diameter of the *i*th cell in the given region, and *n* is the number of cells. The cell density $N_{f}$ could be calculated by the following equation
(2)$$N_{f}={\left(\frac{n\times M^{2}}{A}\right)}^{\frac{3}{2}}$$
where *M* is the magnification of SEM image, and *A* is the area of the SEM picture.

Cells in the parallel section were deformed due to the shear flow, as shown in the Figure 5. The ratio of length to diameter (*d*) was calculated using the equation
(3)$$d\;=\;\frac{L}{B}$$
where *L* is the major axis length of the cells and *B* is the minor axis length of the cells. The angle between the major axis of the cell and melt flow direction is defined as the tilt angle (θ), which is shown in Figure 5. 

The surface quality was observed with the JSM-IT300 (JEOL Ltd., Tokyo, Japan) scanning electron microscope (SEM), and the observation surface was depicted in Figure 4.

## 3. Results and Discussion 

### 3.1. Mechanical Properties

The tensile results of foamed and solid composites with different nano-CaCO_3_ content was shown in Figure 6. It can be clearly seen that the tensile strength of the foamed composites was significantly lower than that of the solid ones. Due to the presence of the cells, the force bearing area of the foamed composites was reduced. As the content of nano-CaCO_3_ increased, the tensile strength of the foamed and solid composites first increased and then decreased. The contact surface of the nano-CaCO_3_ with the polymer matrix was large, and the contact between the nano-CaCO_3_ and the polymer was enhanced with the help of the compatibilizer. As the nano-CaCO_3_ content increased, the interaction between the nano-CaCO_3_ and the PP matrix was further enhanced, thus increasing the strength. However, when an excessive amount of nano-CaCO_3_ was added, the reinforcing effect was lowered, which in turn led to a decrease in strength. The specific tensile strength of foamed and solid composites were also investigated, as shown in Figure 6d. The density of foamed parts with 0, 2, 4, 6, 8, and 10 wt % nano-CaCO_3_ is 0.85, 0.83, 0.83, 0.84, 0.85, and 0.86 g/cm^3^, respectively. It can be found from the figure that the specific tensile strength of foamed composites was higher than that of solid ones. The specific tensile strength of foamed composites increased by 7.8% compared with that of solid composites when adding 6 wt % nano-CaCO_3_. 

The flexural results of foamed and solid composites with different nano-CaCO_3_ content was shown in Figure 7. The flexural strength of solid and foamed composites increased and then decreased with the increase of nano-CaCO_3_ content, and the flexural strength of the foamed and solid composites reached the maximum value when 6 wt % nano-CaCO_3_ was added. The addition of nano-CaCO_3_ could increase the rigidity of the material, and the ability of the material to resist deformation was improved. However, after adding excess nano-CaCO_3_, the agglomerated nano-CaCO_3_ particles reduced the reinforcing effect. The specific flexural strength was also investigated, and the density of foamed parts with 0, 2, 4, 6, 8, and 10 wt % nano-CaCO_3_ is 0.85, 0.88, 0.83, 0.84, 0.85, and 0.86 g/cm^3^, respectively. The changing trend of specific flexural strength was the same as that of the flexural strength. However, due to the lower density of the foamed composites, the specific flexural strength of the foamed composites may be higher than that of the solid ones.

It is well known that cells in a sample significantly affect the impact strength, and small, dense and uniformly distributed cells contribute to the improvement of impact strength. Compared with solid composites, nano-CaCO_3_ had a more pronounced effect on foamed composites. As shown in Figure 8b, with the addition of nano-CaCO_3_, the impact strength of the foamed composites changed drastically, which meant that the cellular structure seriously affected the impact properties. The impact strength of pure foamed sample decreased about 20% compared to that of pure solid sample, while the impact strength of the foamed sample had almost no decrease compared to their counterparts when the nano-CaCO_3_ content was 6 wt %. This is because some larger cells caused local stress concentrations that reduced the impact strength. The density of foamed parts with 0, 2, 4, 6, 8, and 10 wt % nano-CaCO_3_ is 0.85, 0.83, 0.83, 0.84, 0.85, and 0.86 g/cm^3^, respectively, and the change trend of the specific impact strength of the foamed composites was consistent with that of the impact strength. When the nano-CaCO_3_ content reached 4 wt %, the specific impact strength of the foamed composites was higher than that of the solid ones, which indicated that the addition of nano-CaCO_3_ did improve the impact strength of the composites.

### 3.2. Cellular Structure 

The cellular structure in vertical section of the IMD/MIM sample was different from that in parallel section, as shown in Figure 9. The vertical section of IMD/MIM sample was divided into three layers (core layer, transition layer on the film side, and transition layer on the non-film side) according to the cell size distribution, and the parallel section was divided into the same three layers according to the cell degree of deformation. The cell size, cell density, and cell morphology of transition layer on the film side might be different with that of the transition layer on the non-film side. All three layers of foamed composites with different nano-CaCO_3_ contents were investigated.

#### 3.2.1. Cellular Structure in the Vertical Section 

The cellular structure of foamed composites in vertical sections with different nano-CaCO_3_ was shown in Figure 10. It can be found that the cells in the core layer were larger than those in the transition layer. The higher temperature of core layer provided longer growth times for cells. The transition layer cells on the film side were larger than those on the non-film side. The temperature on the film side was higher than that on the non-film side, so the cells had a longer time to grow. Besides, the melt strength on the film side was lower, which made the cells more prone to agglomeration.

The cell size distribution of vertical sections with different nano-CaCO_3_ content was shown in Figure 11. The cell size was distributed between 0 and 120 μm and was mainly concentrated between 20 and 40 μm. This meant that there were many tiny cells in the transition layer, and there were some larger cells in the core layer, which were about 100 μm in size, but the ratio was small. It implied that the cell diameter was opposite to the cell density. It was difficult to analyze the changes of cell diameter and cell density of the whole cross section after adding nano-CaCO_3_. Therefore, we studied the changes of cell density and average cell diameter of different layers relative to the content of nano-CaCO_3_.

The cell density and cell diameter of samples with different nano-CaCO_3_ content were shown in Figure 12. As we can see, the cell density of core layer and transition layers increased firstly and then decreased with the increase of nano-CaCO_3_ content. Nano-CaCO_3_ as a heterogeneous nucleating agent, its addition could reduce the energy barrier of cell nucleation, which helped to improve the nucleation efficiency of the cell. However, excessive nano-CaCO_3_ would agglomerate to form larger CaCO_3_ particles, resulting in reduced nucleation efficiency of cells. The cell density of transition layer on the film side was a little bigger than that on the non-film side. Due to the higher temperature on the film side, the solubility of the gas in the melt was reduced and the thermodynamic instability of the gas was enhanced, thereby increasing the nucleation rate of the cells.

As the nano-CaCO_3_ content increased, the cell diameter of the core layer decreased first and then increased, but the cell diameter of the transition layer did not change much. The transition layer temperature was low and there was not enough time for cells to grow. The cell diameter on the film side was slightly larger than that on the non-film side, this is due to the longer growth time of the cells on the film side and the easier merging of the cells.

#### 3.2.2. Cellular Structure in the Parallel Section

The cellular structure of foamed composites in vertical sections with different nano-CaCO_3_ content was shown in Figure 13. Due to the severe deformation of the cells in parallel sections, we no longer study the cell size, but study the ratio of length to diameter (d) and tilt angle (θ). The d was affected by the cell deformation, and more severe cell deformation corresponded to a larger d. The θ was affected by the melt temperature and viscosity, and higher temperature and lower viscosities meant a larger θ.

During the melt filling stage, the cells of the transition layer were deformed by the action of the shear flow during growth. As the nano-CaCO_3_ content increased, the cell diameter of the core layer decreased first and then increased, while the cell density of the core layer changed in the opposite direction to the cell diameter. The cell diameter of core layer was 72.71 µm and the cell density was 6.37 × 10^9^ cells/cm^3^ when the content of nano-CaCO_3_ was 6 wt %. The addition of nano-CaCO_3_ promoted cell nucleation, providing more nucleation sites and producing more cells. Excessive nano-CaCO_3_ was prone to aggregation, and nucleation sites were reduced, resulting in a decrease in cell density.

The cell density of transition layers was shown in Figure 14b. The cell density of the transition layer changed in the same direction as that of the core layer with the increase of nano-CaCO_3_ content, and the cell density on the film side was larger than that on the non-film side. Higher temperature on the film side increased the thermodynamic instability, thereby increasing the cell nucleation rate. 

The cell tilt angle and the ratio of length to diameter of transition layers were shown in Figure 14c and d, respectively. We can find that θ on the film side was bigger than that on the non-film side, and d was smaller than that on the non-film side. Higher temperature on the film side caused lower viscosity, which meant that cells were less subject to shear, resulting smaller d and smaller angle of the cell from the vertical direction, so θ was larger. As the nano-CaCO_3_ content increased, θ decreased and then increased, and d changed in the opposite direction to θ. The addition of nano-CaCO_3_ increased the number of cell nucleation, resulting in thinner cell walls and stronger cell interactions. With the further addition of nano-CaCO_3_, the melt strength increased and the cell deformation was alleviated.

### 3.3. Surface Quality

The reasons for the formation of surface defects in microcellular injection molded parts were shearing broken cells from melt front and cells spilled from melt surface, as shown in Figure 15. Bubbles that spilled from the surface of the melt had a shorter growth time. It can be clearly seen that the size of cells spilled from melt surface was smaller and the cells were flatted partly by the film as shown in Figure 15a. The cells at the melt front grew during the filling stage, ruptured at the surface. The broken cells at the melt front were large and severely deformed, as shown in Figure 15b, and these cells were difficult to be flattened. In conclusion, the main cause of surface defect formation is the ruptured cells at the melt front.

The surface morphology on the film side of foamed composites with different nano-CaCO_3_ content was shown in Figure 16. It can be seen that although the temperature on the film side was higher, there were still bubble marks left on the surface. With the increase of nano-CaCO_3_ content, the number of bubble marks on the surface increased first and then decreased, that is, the surface quality showed a trend of decreasing first and then increasing. As a heterogeneous nucleating agent, the increase in the content of nano-CaCO_3_ better promotes the heterogeneous nucleation process of the cells, thereby increasing the number of cells generated during the filling process. The deformed cells turn over to the part surface under the effect of fountain flow, so there are more bubble marks. However, with the further increase of the nano-CaCO_3_ content, on the one hand, the agglomeration of the particles leads to a reduction in the number of nucleation points, which leads to a decrease in the nucleation efficiency. On the other hand, the addition of nano-CaCO_3_ will hinder the movement of polymer molecular chains to a certain extent, thereby reducing the melt flow rate and causing the melt strength to increase. Even if the cells reach the part surface, due to the higher melt strength, fewer cells will break and form cell marks, so the surface quality of the part is gradually improved.

## 4. Conclusions

The mechanical properties, cellular structure, and surface quality of PP/nano-CaCO_3_ foamed composites were investigated in combined in-mold decoration and microcellular injection molding (IMD/MIM) process. The results showed that the mechanical properties presented a trend of increasing first and then decreasing with the increase of nano-CaCO_3_ content, and the mechanical properties were optimum when the nano-CaCO_3_ content was 6 wt %. Compared with the solid composites, the foamed composites had poor mechanical properties but better specific mechanical properties. As the blowing agent, the addition of nano-CaCO_3_ inevitably affected the cellular structure. In the vertical section, the cell size and density of the transition layer on the film side were larger than that on the non-film side. As the nano-CaCO_3_ content increased, the cell size of the core layer increased first and then decreased, while the cell density of the core layer changed inversely with the cell size. In the parallel section, the cell ratio of length to diameter on the film side was smaller than that on the non-film side, and the cell tilt angle was larger than that on the non-film side. Changes in cell density and size in the core layer were consistent with those in the vertical section. The broken cells at the melt front were the main cause of surface defects. With the increase of nano-CaCO_3_ content, the surface quality showed a trend of decreasing first and then increasing. The addition of nano-CaCO_3_ increased the number of cell nucleations in the melt filling stage, thereby increasing the bubble marks on the surface, resulting in a decrease in surface quality. With the further addition of nano-CaCO_3_, the melt strength was increased, the cells were not easily broken, and the number of broken cells in the melt front was reduced, thereby improving the surface quality.

## Figures and Tables

**Figure 1 polymers-12-00363-f001:**
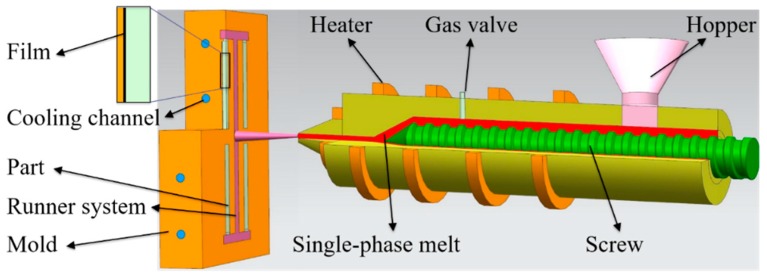
The process of IMD/MIM.

**Figure 2 polymers-12-00363-f002:**
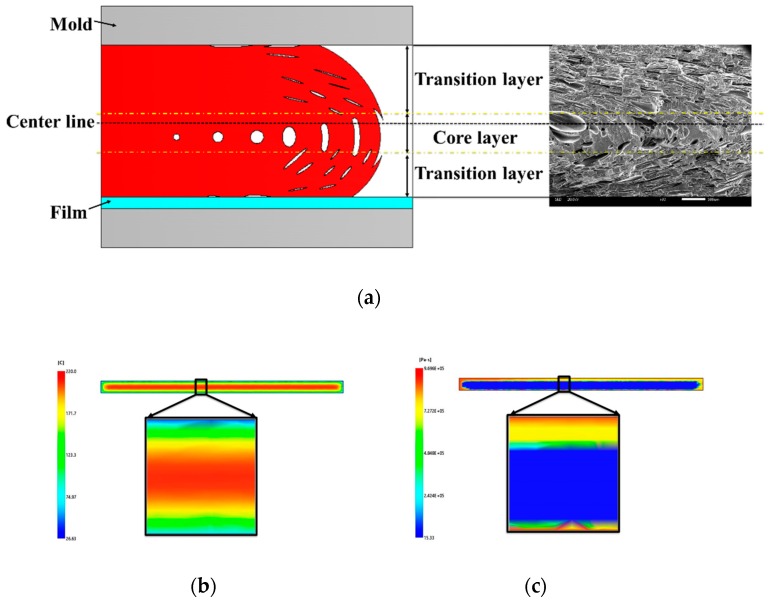
(**a**) Melt flow and cell deformation of IMD/MIM process; (**b**) temperature of parts; (**c**) viscosity of parts.

**Figure 3 polymers-12-00363-f003:**
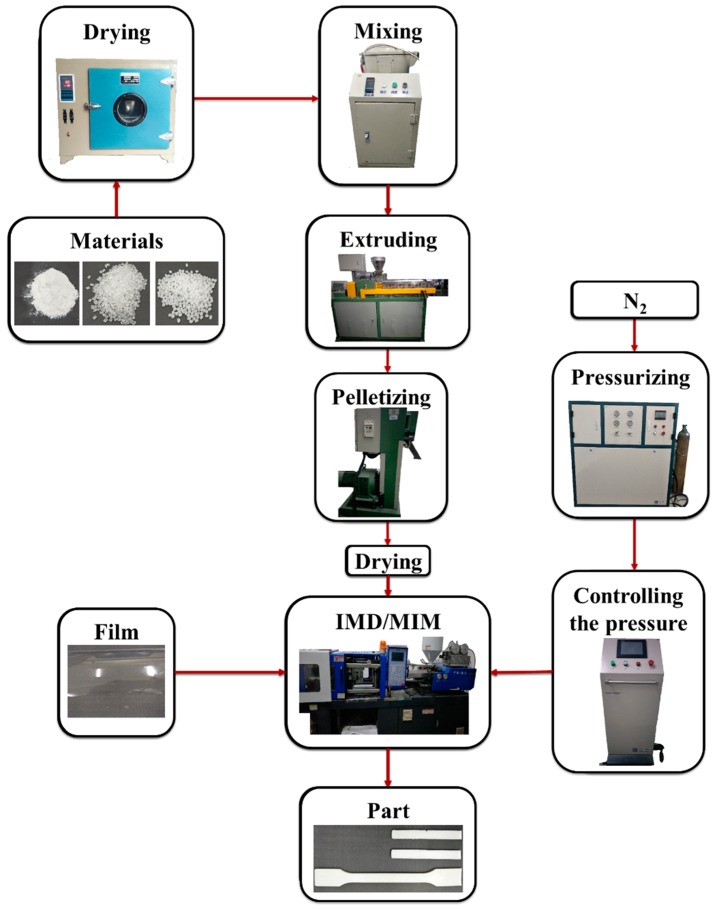
Process flow chart.

**Figure 4 polymers-12-00363-f004:**
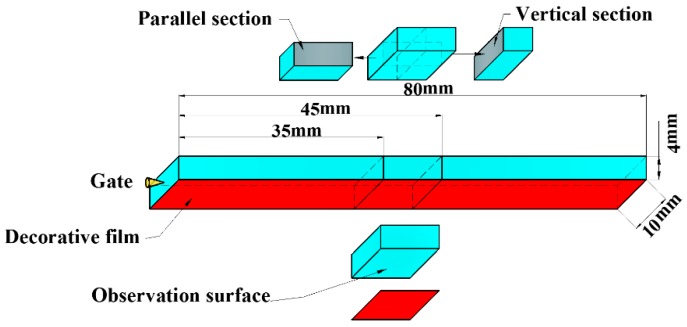
Preparation for vertical and parallel sections and observation surface from flexural sample.

**Figure 5 polymers-12-00363-f005:**
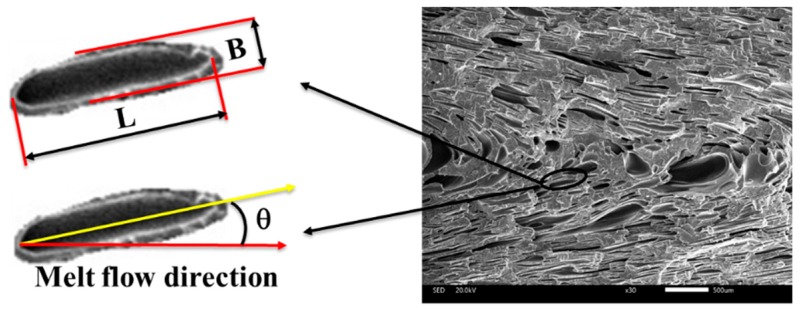
Schematic diagram of ratio of length to diameter ratio and tilt angle.

**Figure 6 polymers-12-00363-f006:**
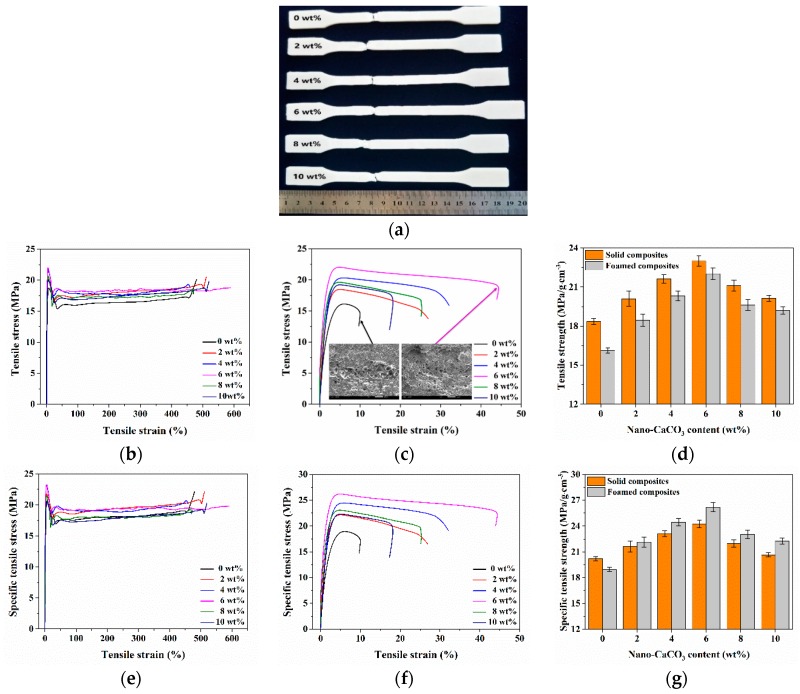
Tensile results of solid and foamed composites: (**a**) foamed samples after tensile tests; (**b**,**c**) tensile stress strain curves of solid and foamed composites, respectively; (**d**) the tensile strength; (**e**,**f**) specific tensile stress strain curves of solid and foamed composites, respectively; (**g**) the specific tensile strength.

**Figure 7 polymers-12-00363-f007:**
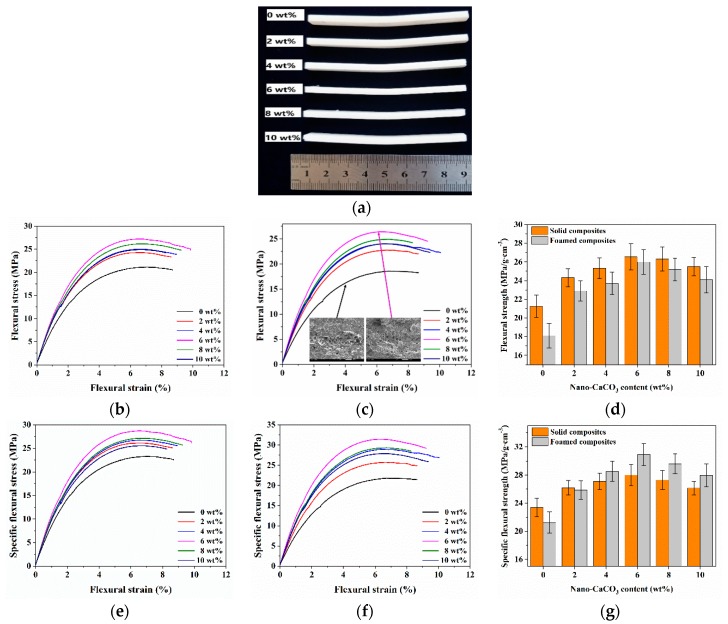
Flexural results of solid and foamed composites: (**a**) foamed samples after flexural tests; (**b**,**c**) flexural stress strain curves of solid and foamed composites, respectively; (**d**) the flexural strength; (**e**,**f**) specific flexural stress strain curves of solid and foamed composites, respectively; (**g**) the specific flexural strength.

**Figure 8 polymers-12-00363-f008:**
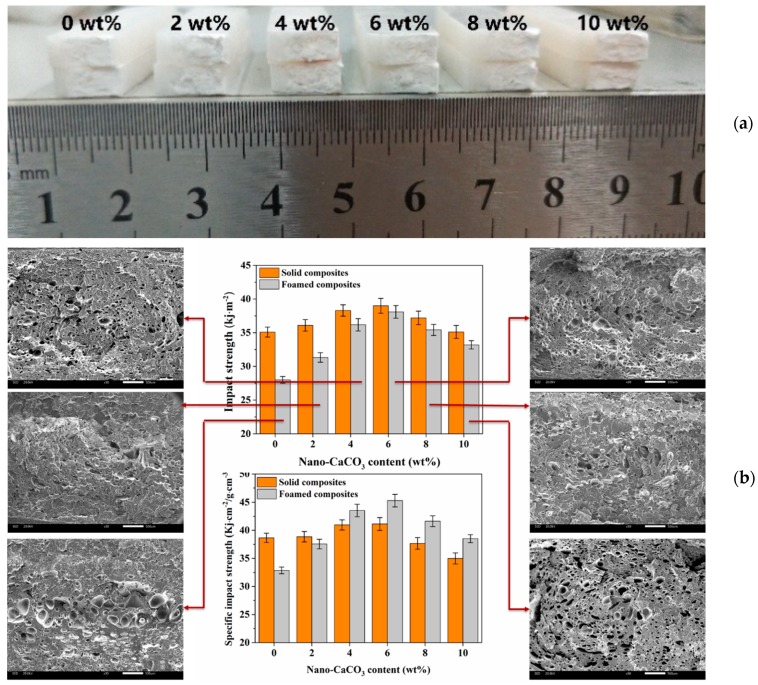
Impact results of solid and foamed composites: (**a**) samples after impact tests; (**b**) the impact strength and specific impact strength.

**Figure 9 polymers-12-00363-f009:**
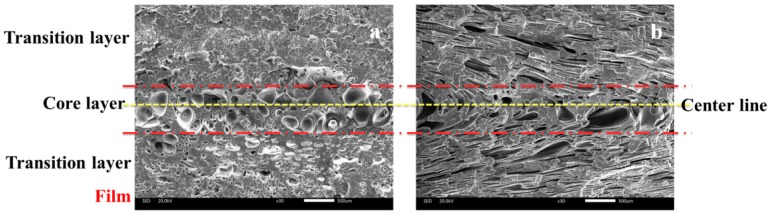
The cellular structure in vertical and parallel sections of IMD/MIM sample: (**a**) vertical section; (**b**) parallel section.

**Figure 10 polymers-12-00363-f010:**
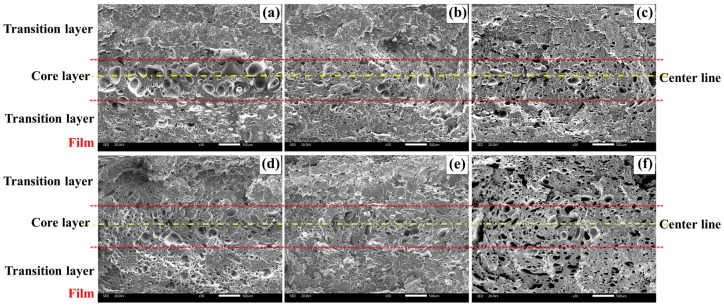
Cellular structure of foamed composites in vertical sections with different nano-CaCO_3_ content: (**a**) 0 wt %; (**b**) 2 wt %; (**c**) 4 wt %; (**d**) 6 wt %; (**e**) 8 wt %; (**f**) 10 wt %.

**Figure 11 polymers-12-00363-f011:**
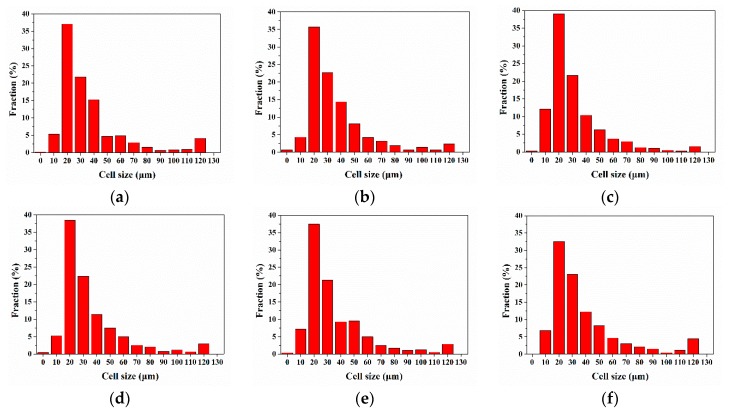
Cell size distribution of vertical sections with different nano-CaCO_3_ content: (**a**) 0 wt %; (**b**) 2 wt %; (**c**) 4 wt %; (**d**) 6 wt %; (**e**) 8 wt %; (**f**) 10 wt %.

**Figure 12 polymers-12-00363-f012:**
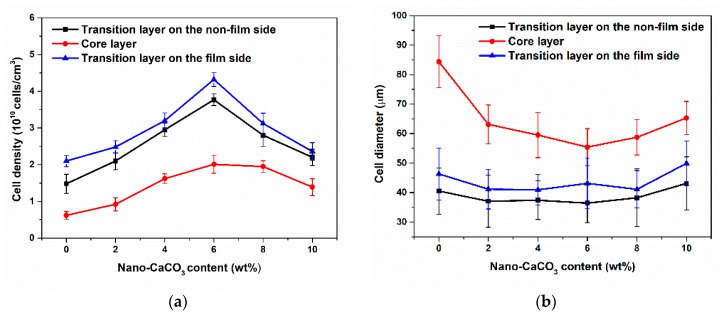
The cell density and average cell diameter of samples with different nano-CaCO_3_ content: (**a**) cell density; (**b**) cell diameter.

**Figure 13 polymers-12-00363-f013:**
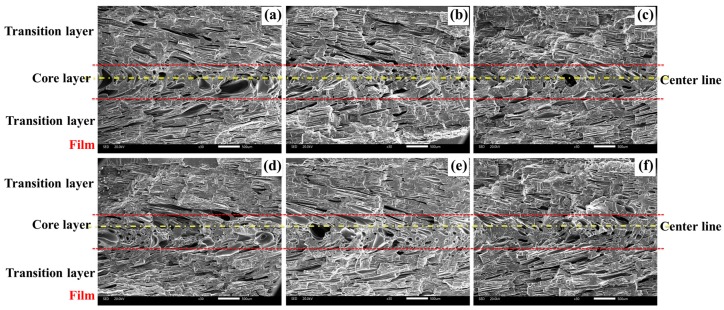
Cellular structure of foamed composites in parallel sections with different nano-CaCO_3_ content: (**a**) 0 wt %; (**b**) 2 wt %; (**c**) 4 wt %; (**d**) 6 wt %; (**e**) 8 wt %; (**f**) 10 wt %.

**Figure 14 polymers-12-00363-f014:**
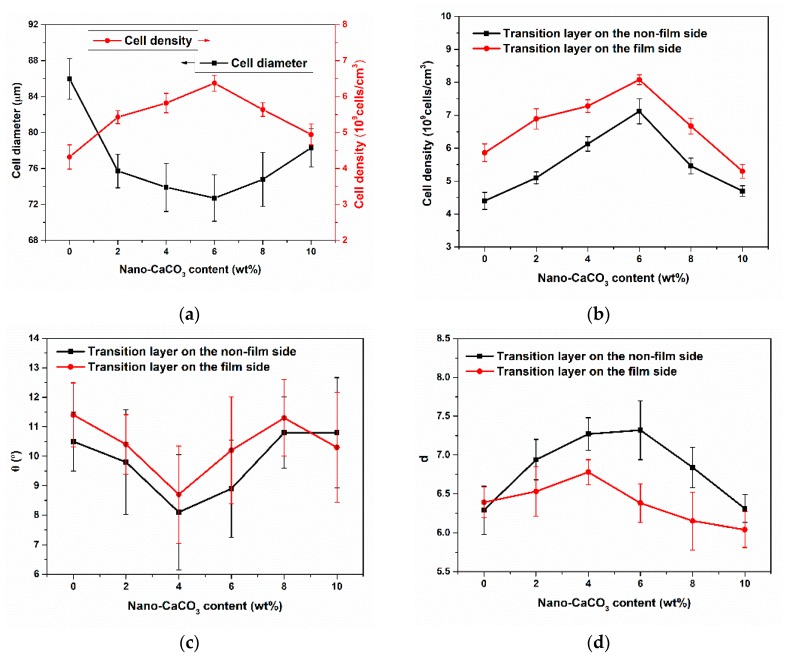
(**a**) Cell diameter and cell density of core layer; (**b**) cell density of transition layers; (**c**) cell tilt angle of transition layers; (**d**) cell ratio of length to diameter of transition layers.

**Figure 15 polymers-12-00363-f015:**
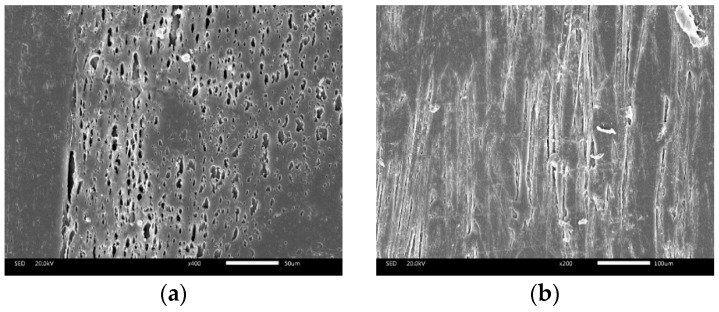
Different types of cells on the surface: (**a**) cells spilled from melt surface; (**b**) broken cells from melt flow front.

**Figure 16 polymers-12-00363-f016:**
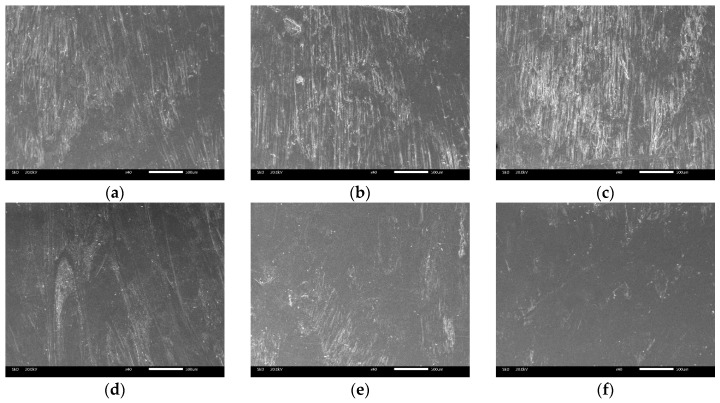
The surface morphology on the film side of foamed composites with different nano-CaCO_3_ content: (**a**) 0 wt %; (**b**) 2 wt %; (**c**) 4 wt %; (**d**) 6 wt %; (**e**) 8 wt %; (**f**) 10 wt %.

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
