# Peer review of "Investigation on Foamed PP/Nano-CaCO3 Composites in a Combined in-Mold Decoration and Microcellular Injection Molding Process"

_polymers, 2020, doi:10.3390/polym12020363_

Round 1
Reviewer 1 Report
The authors prepared PP/Nano-CaCO3 composites using in-mold decoration and microcellular injection molding method. Some information and explanation of this study are still lacked.
1. The similar investigation already published from the same and the other groups. This manuscript should check the self-plagiarism (Polymers, 2019, 11, 778; doi:10.3390/polym11050778). The authors may organize the results and use the new concept to convince readers why they should refer and read this study.
2. The authors mentioned, “They found that with 5 wt% nano-CaCO3 addition foamed composites produced optimum cell structure. Mao investigated the content of nano-CaCO3 on mechanical properties, and the cell structure in different sections, and it was found that with the 6 wt% addition of nano-CaCO3, the mechanical properties was optimum [22].” The variation of this manuscript is the different content of nano-CaCO3. The optimized addition of nano-CaCO3 has been measured from the other teams. In this manuscript, the authors mentioned that the niche is improvement the surface quality of foamed parts. From figure 16, I cannot distinguish the “improvement the surface quality of foamed parts”. How important is the surface quality of foam? The niche of this manuscript may be weak for readers. Please compare the results with other references.
3. Please add the method of specific tensile strength in 2. Experimental.
4. Please show the statistical results of all strength data.
5. In figure 12, there is no standard deviation.
6. Line 142, vertical section (10×4 “mm”). Please check the unit.
7. Line 284, density was 6.37×109 cells/cm3. Please check the superscript.
Author Response
Dear reviewer:
We sincerely thank you for your careful reading and pertinent advices. We have modified the manuscript according to your suggestions, all revised part of manuscript was marked in red and detailed corrections are listed below:
The similar investigation already published from the same and the other groups. This manuscript should check the self-plagiarism (Polymers, 2019, 11, 778; doi:10.3390/polym11050778). The authors may organize the results and use the new concept to convince readers why they should refer and read this study.Response: Thanks for your suggestion. The characterization methods of the two papers are indeed similar, because the characterization methods commonly used in this research area are mainly crystallization, cellular structure, mechanical properties, and surface quality. In addition, the two papers use different processes, and the differences in processes will have a significant impact on the characterization results. The previous paper used the microcellular injection molding process, while this article uses the combined in-mold decoration and microcellular injection molding process. Compared with the previous paper, the paper reduces the heat transfer coefficient on the film side due to the existence of the film, which causes the temperature of the melt to be asymmetric in the thickness direction, which results in asymmetric melt viscosity and asymmetry flow. The asymmetry temperature causes the cells to shift to the film side, and the changes in the cellular structure further affect the mechanical properties. At the same time, the existence of the film makes the melt temperature on the film side to be higher, and the bubble marks turned over to the surface of the part are dissolved back into the melt, thereby reducing the bubble marks on the surface and improving the surface quality of the part. Therefore, compared with the previous paper, this article adds a study of surface quality. In summary, this article has some research value.
The authors mentioned, “They found that with 5 wt% nano-CaCO3 addition foamed composites produced optimum cell structure. Mao investigated the content of nano-CaCO3 on mechanical properties, and the cell structure in different sections, and it was found that with the 6 wt% addition of nano-CaCO3, the mechanical properties was optimum [22].” The variation of this manuscript is the different content of nano-CaCO3. The optimized addition of nano-CaCO3 has been measured from the other teams. In this manuscript, the authors mentioned that the niche is improvement the surface quality of foamed parts. From figure 16, I cannot distinguish the “improvement the surface quality of foamed parts”. How important is the surface quality of foam? The niche of this manuscript may be weak for readers. Please compare the results with other references.
Response: Thanks for your suggestion. The two references are different in the process of studying the effect of nano-CaCO3 content on the cell structure, and the differences in processes will have a significant impact on the characterization results. In continuous foaming, the cell structure is optimal when 5% nano-CaCO3 is added, but in intermittent foaming, the cell structure is optimal when 6% nano-CaCO3 is added. The bubble marks formed by the cracked and deformed cells left on the part surface under the flattening of the mold seriously affect the part surface quality, which greatly limits the widespread application and promotion of MIM. Therefore, it is extremely important to study how to improve the surface quality of the part. For this purpose, we adopt combined in-mold decoration and microcellular injection molding process. The existence of the film makes the melt temperature on the film side to be higher, and the bubble marks turned over to the surface of the part are dissolved back into the melt, thereby reducing the bubble marks on the surface and improving the part surface quality. As for Figure 16, we judge the surface quality based on the number of bubble marks on part surface. Less bubble marks on part surface means better surface quality. It can be clearly seen that with the increase of Nano-CaCO3 content, the bubble marks on part surface first increase and then decrease, which indicates that the surface quality of the part becomes worse first and then becomes better.
Please add the method of specific tensile strength in 2. Experimental.
Response: Thanks for your suggestion. The method of specific strength has been added to the characterization of this paper.
Please show the statistical results of all strength data.
Response: Thanks for your suggestion. The statistical results of all strength data have been provided.
In figure 12, there is no standard deviation.
Response: Thanks for your suggestion. The standard deviation of Fig. 12 has been added, and the standard deviation of Fig. 14 has also been added.
Line 142, vertical section (10×4 “mm”). Please check the unit.
Response: Thanks for your suggestion. The unit mm has been modified to mm2.
Line 284, density was 6.37×109 cells/cm3. Please check the superscript.
Response: Thanks for your suggestion. 6.37×109 cells/cm3 has been modified to 6.37×109 cells/cm3. In response to this error, I have carefully checked the entire paper to ensure that it is correct.

Reviewer 2 Report
General comments
Authors just presented initial data. More detailed analysis required. Besides, manuscript is very badly written and littered with many inappropriate and meaningless references. Requires major revision.
Specific comments
Introduction
Lines #34-35: Poor quality of writing. "Especially in terms of surface quality, the surface quality decreases sharply due..." The phrase "surface quality" repeating, rephrase this statement.
Lines#42-43. Many errors in formatting of manuscript. For example: In line #43: "stage [11-13]"
Lines 48-49. Confusing statement "Microcellular co-injection molding is injecting solid melt before injecting melt with supercritical fluid (SCF),.." What is "injecting solid melt before injecting melt". Rephrase.
Lines #53. Authors state "Jungjoo et al. investigated.." but no reference added.
In Line #56, authors state "Ding et al. investigated the effects of nano-CaCO3.." and cite papers published in 2013 to support. Do authors seriously think that until 2013, no one ever used CaCO3 in foams and Ding was the first to use CaCO3 as filler!! Remove these inappropriate references.
Lines #61 to #64. Remove reference #23 and its corresponding explanation. It has nothing to do with PP/CaCO3.
Lines #64-67. You cannot compare nano-clay, a 2-D filler with a spherical CaCO3 filler. The reinforcing mechanism and even the foaming behaviour is also different. Remove References 24 and 25 or give proper scientific correlation.
Line #76. No need for "respectively"
Quality of Figure 2, especially the scale bar in 2(b) and (c) is of very poor quality. Increase the font size of scale bar. Quality of SEM in 2(a) is also very poor. The scale is not visible.
Secondly why PET film only on one side. Why not adher PET film on both sides. Justify this anomaly in experimentation.
Thirdly the schematic diagram in 2(a) should be backed up with detailed morphological analysis by SEM. Providing one SEM and that too of very poor quality with almost no foam will not suffice. A detailed "number of cells per unit volume", "average,minimum, and maximum cell size" etc. should be carried out.
There are many more problems with this manuscript. But above are enough to recommend, major revision
Author Response
Dear reviewer:
We sincerely thank you for your careful reading and pertinent advices. We have modified the manuscript according to your suggestions, all revised part of manuscript was marked in red and detailed corrections are listed below:
Lines #34-35: Poor quality of writing. "Especially in terms of surface quality, the surface quality decreases sharply due..." The phrase "surface quality" repeating, rephrase this statement.Response: Thanks for your suggestion. The statement has been rephased. “the surface quality decreases sharply” has been changed to “the appearance of the part is poor”.
Lines#42-43. Many errors in formatting of manuscript. For example: In line #43: "stage [11-13]"
Response: Thanks for your suggestion. After careful inspection, the format has been changed.
Lines 48-49. Confusing statement "Microcellular co-injection molding is injecting solid melt before injecting melt with supercritical fluid (SCF),.." What is "injecting solid melt before injecting melt". Rephrase.
Response: Thanks for your suggestion. The statement has been rephrased.
Lines #53. Authors state "Jungjoo et al. investigated.." but no reference added.
Response: Thanks for your suggestion. The reference has been added.
In Line #56, authors state "Ding et al. investigated the effects of nano-CaCO3.." and cite papers published in 2013 to support. Do authors seriously think that until 2013, no one ever used CaCO3 in foams and Ding was the first to use CaCO3 as filler!! Remove these inappropriate references.
Response: Thanks for your suggestion. Citing this reference does not imply that Ding was the first person to introduce CaCO3 during the microcellular foaming process, but it is more persuasive to cite references from newer literature. A 2007 related literature has been added to this paper (H. Huang, J. Wang, Improving polypropylene microcellular foaming through blending and the addition of nano-calcium carbonate, J. Appl. Polym. Sci. 106 (2007) 505-513.).
Lines #61 to #64. Remove reference #23 and its corresponding explanation. It has nothing to do with PP/CaCO3.
Response: Thanks for your suggestion. The reference and its corresponding explanation have been removed.
Lines #64-67. You cannot compare nano-clay, a 2-D filler with a spherical CaCO3 filler. The reinforcing mechanism and even the foaming behaviour is also different. Remove References 24 and 25 or give proper scientific correlation.
Response: Thanks for your suggestion. The references 24 and 25 and their corresponding explanation have been removed.
Line #76. No need for "respectively"
Response: Thanks for your suggestion. “respectively” has been removed.
Quality of Figure 2, especially the scale bar in 2(b) and (c) is of very poor quality. Increase the font size of scale bar. Quality of SEM in 2(a) is also very poor. The scale is not visible.
Response: Thanks for your suggestion. Fig. 2 has been modified, and now the scale bar is visible as well as the SEM photo.
Secondly why PET film only on one side. Why not adhere PET film on both sides. Justify this anomaly in experimentation.
Response: Thanks for your suggestion. Because only one side of the part has surface quality requirements and the other side serves as a connection. Therefore, only one side of the part needs to be filmed, not two sides.
Thirdly the schematic diagram in 2(a) should be backed up with detailed morphological analysis by SEM. Providing one SEM and that too of very poor quality with almost no foam will not suffice. A detailed "number of cells per unit volume", "average, minimum, and maximum cell size" etc. should be carried out.
Response: Thanks for your suggestion. The number of cells in the SEM image in Fig. 2(a) is not small, but because the transition layer cells are severely deformed into ellipse shapes under the action of shear flow, so the cell quality is poor. This SEM picture is placed here to illustrate more clearly under the conditions of combined in-mold decoration and microcellular injection molding process conditions, due to the existence of the film, the heat transfer coefficient on the film side is reduced, resulting in a higher temperature on the film side and causing cells to shift to the film side. Therefore, the cells in this SEM image have not been quantitatively analyzed, which will be analyzed in detail in the subsequent results and discussion section.

Round 2
Reviewer 1 Report
From the response, I did not see a new concept (Question 1) and statistical results (question 4) in the new manuscript. There is no standard deviation in figure 12a. (Question 5)
Author Response
Dear reviewer:
We sincerely thank you for your careful reading and pertinent advices. We have modified the manuscript according to your suggestions, all revised part of manuscript was marked in red and detailed corrections are listed below:
Regarding question 1, compared with the previous microcellular injection molding, this article used a combined in-mold decoration and microcellular injection molding process. The heat transfer coefficient on the film side due to the existence of the film is reduced, which causes the temperature of the melt to be asymmetric in the thickness direction, which results in asymmetric melt viscosity and asymmetry flow. The asymmetry temperature causes the cells to shift to the film side. These phenomena are described in section 2.1. In the results and discussion, since the foamed parts achieve weight reduction, the analysis of specific strength is more convincing than the strength, so the mechanical properties in section 3.1 increases the comparative analysis of specific strength. Compared with the traditional microcellular injection molding, due to the existence of the film, the cells are no longer symmetrically distributed, but are offset to a certain distance from the film side. Therefore, a detailed comparison and analysis of the cells with and without the film-side transition layer is performed in section 3.2. In addition, the existence of the film makes the melt temperature on the film side to be higher, and the bubble marks turned over to the surface of the part are dissolved back into the melt, thereby reducing the bubble marks on the surface and improving the surface quality of the part. Therefore, research on surface quality has been added, which is reflected in section 3.3.
Regarding question 4, the statistical results of all strength data have been shown in section 3.1 of this paper.
For question 5, regarding the cell density, we calculated the number of cells based on a SEM photo and then calculated it according to Eq. 2. Since the cell density is not an average value, there is no standard deviation in Fig. 12 (a).
